# Work-related psychosocial risk factors for stress-related mental disorders: an updated systematic review and meta-analysis

Henk F van der Molen [ID] , Karen Nieuwenhuijsen, Monique H W Frings-Dresen, Gerda de Groene

**To cite:** van der Molen HF, Nieuwenhuijsen K, Frings-Dresen MHW, *et al.* Work-related psychosocial risk factors for stress-related mental disorders: an updated systematic review and meta-analysis. *BMJ Open* 2020;**10**:e034849. doi:10.1136/bmjopen-2019-034849

HFvdM and GdG contributed equally.

Coronel Institute of Occupational Health, Amsterdam UMC—Locatie AMC, Amsterdam, North Holland, The Netherlands

**Correspondence to**
Dr Henk F van der Molen;
h.f.vandermolen@amsterdamumc.nl

## ABSTRACT

**Objective** The objective was to conduct an update of a previously published review and meta-analysis on the association between work-related psychosocial risk factors and stress-related mental disorders (SRD).

**Design** Systematic review and meta-analysis.

**Data sources** Medline, Embase and PsycINFO were searched for articles published between 2008 and 12 August 2019 and references of a systematic review performed for the period before 2008 were included. Primary prospective studies were included when outcome data were described in terms of SRD assessment or a dichotomous outcome, based on a validated questionnaire, and at least two levels of work-related exposure were reported (exposed vs less or non-exposed). We used GRADE to assess the evidence for the associations between risk factors and the onset of SRD.

**Results** Seventeen studies met the inclusion criteria. In total, a population of 73 874 workers from Belgium, Denmark, England, Finland, Japan, the Netherlands and Sweden were included in the meta-analysis of 14 prospective cohort studies. This meta-analysis revealed moderate evidence for associations between SRD and effort reward imbalance (OR=1.9, 95% CI 1.70 to 2.15), high job demands (OR=1.6, 95% CI 1.41 to 1.72), organisational justice (ORs=1.6 to 1.7, CIs 1.44 to 1.86), social support (ORs=1.3 to 1.4, CIs 1.16 to 1.69), high emotional demands (OR=1.6, 95% CI 1.35 to 1.84) and decision authority (OR=1.3, CI 1.20 to 1.49). No significant or inconsistent associations were found for job insecurity, decision latitude, skill discretion and bullying.

**Conclusion** Moderate evidence was found that work-related psychosocial risk factors are associated with a higher risk of SRD. Effort-reward imbalance, low organisational justice and high job demands exhibited the largest increased risk of SRD, varying from 60% to 90%.

### Strengths and limitations of this study

► This systematic review, including a meta-analysis of prospective cohort studies, gives a valuable overview in which work-related psychosocial risk factors increase the risk of stress-related mental disorders.

► The case definitions of stress-related mental disorders used validated questionnaires and the methodological quality of the included studies was high.

► The GRADE framework made it possible to assess to the quality of evidence; however, the starting level for grading this evidence for prospective aetiological studies is subject to scientific debate.

► Limitations are the lack of harmonised assessments of stress-related mental disorders and detailed exposure assessment in the identified observational cohort studies.

exhaustion[2] and a prevalence of distress symptoms of up to 50% in specific professions and countries.[3] SRD represent a significant part of work-related common mental disorders in both self-report surveys and notification systems for occupational diseases.[4] The WHO has recently announced[5] that burn-out is included as 'an occupational phenomenon' in the 11th revision of the International Classification of Diseases (ICD-11).

In many countries, SRD are not notified as occupational diseases,[6] mainly because of their multifactorial origin[7] or discussion about the medical status (as with burn-out).[589] Evidence-based knowledge about work-related factors could be used for facilitating a decision on the 'work-relatedness' of an individual case of SRD in a reporting scheme or concerning the selection and implementation of preventive workplace interventions around factors with the strongest effect on SRD.[7]

In the Netherlands, SRD cover diagnoses of adjustment disorders (ICD-10: F43.2) and burn-out as a state of exhaustion (ICD-10:

## INTRODUCTION

Stress-related mental disorders (SRD) are frequently reported in the working population, with varying incidence rates of 13% for psychological distress,[1] 22% for emotional

Z730).[4] These diagnoses differ from just stress symptoms since they are clinically assessed and have in common that they reflect a status where subjective and emotional disturbance interfere with social functioning and performance, arising from a period of adaptation to stressful conditions. Symptoms are what a worker experiences, and these symptoms might negatively affect normal or regular functioning of a person—a so-called disorder. To distinguish between symptoms and disorders, we considered a SRD as diagnosis following a clinical anamnesis, stress complaints as assessed in a score above a cut-off point on a validated questionnaire or absenteeism from work due to stress problems. Work-related psychosocial risk factors can contribute to the onset of SRD.[10]

The high and increasing burden of SRD strengthens the urgency of up-to-date insight into work-related psychosocial risk factors associated with SRD and based on longitudinal study designs in order to initiate relevant primary and secondary preventive interventions at worksites. Previous systematic reviews aiming to determine work-related risk factors for SRD have either not been updated since 2008,[10] combined various common mental disorders, such as anxiety, depression and stress-related disorders, as outcome measures[11] or used other outcomes, such as suicide.[12] Based on seven prospective studies, Nieuwenhuijsen et al[10] reported that high job demands, low job control, low coworker support, low supervisor support, low procedural justice, low relational justice, high effort–reward imbalance and job insecurity (only for men) predicted the occurrence of SRDs. Our aim is to update this previously published review and meta-analysis from 2008[10] to examine: (1) which work-related psychosocial risk factors contribute to the onset of SRD and (2) to what extent these risk factors are associated with SRD.

## METHODS
This review followed the PRISMA Statement (see online supplementary appendix). This review was not registered in Prospero, but an update of a previous review by Nieuwenhuijsen et al.[10]

### Patient and public involvement
There was no patient or public involvement in designing the study, but the systematic review is a response to priority setting in collaboration with policy-makers of the Dutch Ministry of Employment and Social Affairs.

### Study selection
#### Eligibility criteria
Primary prospective cohort studies were included where outcome data were described in terms of clinically or questionnaire-assessed stress-related disorder (present or not) and at least two levels of work-related exposure (exposed vs less or non-exposed) among a working population were reported in order to be able to retrieve or calculate a risk estimate (OR, relative risk (RR) or HR).

This updated systematic review[10] considered studies to have included SRD if the outcome was either (i) a SRD diagnosis following a clinical anamnesis, (ii) a high level of stress complaints as assessed in a score above a cut-off point on a validated questionnaire for fatigue, stress or non-specific mental ill-health or (iii) absenteeism from work due to stress problems. Based on findings of Nieuwenhuijsen et al[10] and identified factors stemming from theoretical stress models, we defined seven types of exposure (i) job demands, (ii) job control (decision latitude, decision authority, skill discretion), (iii) social support (coworker, supervisor, both), (iv) emotional demands, (v) organisational justice (procedural and relational), (vi) effort-reward imbalance and (vii) other psychosocial risk factors, for example, bullying.

Studies that described work-related risk factors in terms of psychosocial risk factors were eligible for inclusion. The studies had to describe workers in a real workplace setting. All types of exposure assessment were eligible for inclusion: self-reports or researcher observations or direct measurements. No additional criteria were formulated regarding latency between exposure and the presence or onset of the disorder or adjustment for confounders.

### Data sources and search terms
We searched the electronic databases of Medline, Embase and PsychInfo for studies between 2008 and 12 August 2019 as described in online supplementary appendix 1. Our PICO can be stated as: p=working population, I/C exposed/less or none exposed to a priori defined exposure categories, O=stress related disorder. Eligible studies before 2008 were retrieved from the systematic review by Nieuwenhuijsen et al.[10]

### Data collection and analyses
#### Study selection process
Titles and abstracts were independently screened by two reviewers to identify potentially relevant studies. We used a free online software tool to screen and assess references (https://rayyan.qcri.org/welcome). The full texts of potentially relevant articles were assessed for eligibility against the inclusion criteria. Disagreement between review authors on the selection of studies for inclusion occurred in less than 5% of the references screened and was resolved by discussion.

#### Data extraction and management
Data were extracted by one review author (GG) and checked by another review author (HM). Data on the following were extracted from each article: author, country of study, study design, case definition of stress-related disorder, sources and number of participants, exposure definition, exposure assessment, exposure categories, risk estimate and adjustment for confounders.

#### Methodological quality assessment
For the assessment of the risk of bias within studies, the quality criteria from the systematic review by Nieuwenhuijsen et al[10] were used for all study designs, with nine items

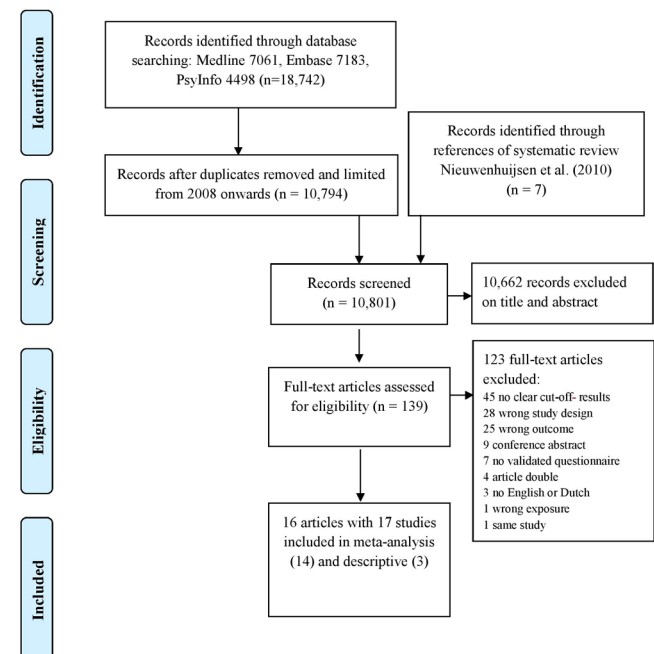

**Figure 1** Prisma flow diagram.

covering the selection of participants, measurement of variables and control for confounding. The quality of the studies was independently assessed and agreed by two review authors (GG, HM). In total, nine items across five categories were evaluated for quality assessment: (1) the main features of the study population were stated, for example, sex, age and work context; (2) the participation rate at baseline was at least 50%; (3) the response at follow-up was at least 70% or the non-response was not selective; (4) the data on psychosocial work factors were collected using standardised instruments; (5) the data on outcome were collected at least at three different time points; (6) the statistical model used was appropriate for the outcome studied; (7) the measures of association with the statistical model were presented; (8) there were study controls for confounding with rationale; (9) the number of cases was at least 10 times the number of independent variables in the analysis. The criteria for each item were scored with 'positive', 'negative' or 'not clear'. A higher quality of a study was defined as ≥5 items scored as 'positive' out of the nine quality criteria.

### Data synthesis
A descriptive analysis of all studies was performed, summarised, classified in categories of psychosocial risk factors and assessed for methodological quality. Risk estimates and the corresponding 95% CI of the association between work-related factors and SRD were extracted and summarised. After assessing probable heterogeneity of risk factor and outcome assessment, all authors discussed and decided on the risk estimates to be included in the meta-analysis or should be analysed separately. We used the rule at least two studies when no statistical heterogeneity ($I^2 < 50\%$) and no heterogeneity in measures of risk factors and stress-related disorders (as discussed among the authors) was found.

### Meta-analyses and quality of evidence
The selection of the work-related risk factors in the meta-analyses was based on (1) sufficient contrast between reported exposure categories, that is, low vs high exposure; (2) effect estimates controlled for other non-work-related factors, as reported in the primary studies; (3) homogeneity in definition or measurement of risk factors.

The meta-analysis was performed in line with the procedure described in Watanabe *et al*[13] by using a log-transformed OR and its SE in Review Manager (Cochrane Review Manager V.5.3). For the main analysis, the main ORs and the SEs from selected studies were subjected to a random-effects model meta-analysis to estimate a pooled OR and its 95% CI.

The quality of evidence—also taken into account the risk of bias across studies—was assessed using the GRADE (Grades of Recommendations, Assessment, Development and Evaluation) framework for prognostic studies developed by Huguet *et al*.[14] Following Huguet *et al*,[14] our starting point for the quality of the evidence was 'moderate' for prospective cohort studies that aimed to identify associations between potential prognostic factors (in our review risk factors) and the outcome. The evidence could decrease on the basis of five factors: study limitations, inconsistency, indirectness, imprecision and publication bias. Moreover, two factors, (1) study findings with moderate or large effect sizes (ie, lower limit of 95% CI OR≥2.0) or (2) an exposure-response gradient, could lead to an upgrade of the quality of evidence. Four levels of quality were used: high, moderate, low and very low.

### RESULTS
#### Selected studies
A PRISMA flow diagram of the study selection process is shown in figure 1. After excluding duplicates, 10 801 references were retrieved from the databases and the systematic review by Nieuwenhuijsen *et al*,[10] and assessed based on title and abstract. The full texts of 139 potentially eligible articles were then examined, of which 17 studies from 16 articles met the inclusion criteria.

#### Psychosocial risk factors and SRD
In total, 17 longitudinal studies in 16 articles[1–3 15–27] described the association between exposure to psychosocial risk factors and the occurrence of assessed mental SRD. The characteristics of the outcome definition (SRD), exposure definition and reported associations are presented in table 1.

#### Methodological quality
The methodological quality of the studies of risk factors varied from scoring 5 out of 9 items to 8 out of 9 items (see table 2). The most frequently missing quality items were response at follow-up less 70%, data on outcome were collected at least at three different time points and study controls for confounding with rationale.

**Table 1** Associations between psychosocial risk factors at work and the occurrence of mental stress related disorders (17 studies)

| Author, country population | Outcome | Exposure | | |
|---|---|---|---|---|
| | Definition and assessment | Definition and assessment | OR | (95% CI) |
| Mino (1999), Japan, machine production, n=310[15] | GHQ-30 GHQ-30≥8 | Job demands: one item | ♂ RR 1.39 | 0.92 to 2.10 |
| | | (always/sometimes present vs absent) | ♀ RR 1.14 | 0.81 to 1.59 |
| | | Supervisor support: one item | ♂ RR 1.10 | 0.69 to 1.74 |
| | | (absent vs always/sometimes present) | ♀ RR 2.21 | 1.25 to 3.89 |
| Stansfeld (1999), England, civil servants, n=10 308[16] | GHQ-30 GHQ-30>4 | Job demands: adapted JCQ | ♂ 1.33 | 1.1 to 1.6 |
| | | (highest tertile vs lowest) | ♀ 1.24 | 1.0 to 1.6 |
| | | Decision authority: adapted JCQ | ♂ 1.29 | 1.1 to 1.5 |
| | | (lowest vs highest tertile) | ♀ 1.37 | 1.1 to 1.8 |
| | | Coworker support: adapted JCQ | ♂ 1.29 | 1.1 to 1.5 |
| | | (lowest vs highest tertile) | ♀ 1.12 | 0.9 to 1.4 |
| | | Supervisor support: adapted JCQ | ♂ 1.31 | 1.1 to 1.5 |
| | | (lowest vs highest tertile) | ♀ 1.11 | 0.9 to 1.3 |
| | | Effort-reward imbalance: imbalance: indicator of high effort and low rewards | ♂ 2.57 | 1.8 to 3.6 |
| | | (high efforts/low rewards vs no high efforts/lowrewards) | ♀ 1.67 | 1.0 to 2.9 |
| | | Skill discretion: adapted JCQ | ♂ 1.11 | 0.9 to 1.3 |
| | | (lowest vs highest tertile) | ♀ 1.09 | 0.8 to 1.4 |
| Bültmann (2002), Netherlands, 45 companies, n=8833[1] | GHQ-12≥4 | Job demands Questionnaire (JCQ) | | |
| | | Job demands: JCQ | ♂ 1.51 | 1.23 to 1.85 |
| | | (highest vs lowest tertie) | ♀ 1.44 | 1.03 to 2.01 |
| | | Decision latitude: JCQ | ♂ 1.14 | 0.9 to 1.43 |
| | | (lowest vs highest tertile) | ♀ 0.88 | 0.62 to 1.24 |
| | | Coworker support: JCQ | ♂ 1.25 | 1.04 to 1.49 |
| | | (low vs high) | ♀ 1.31 | 0.97 to 1.78 |
| | | Supervisor support: JCQ | ♂ 1.25 | 1.05 to 1.49 |
| | | (low vs high) | ♀ 1.12 | 0.85 to 1.47 |
| | | Emotional demands: Dutch QPJW | ♂ 1.73 | 1.40 to 2.14 |
| | | Work and Health,self-formulated (high (2–5) vs no (0,1)) | ♀ 1.39 | 1.01 to 1.91 |
| | | Job insecurity: QPJW, one-item | ♂ 1.63 | 1.18 to 2.27 |
| | | (yes vs no) | ♀ 0.94 | 0.56 to 1.59 |
| Bonde (2005), Denmark, various workplaces, n=2846[17] | SSPI≥4 | Skill discretion: Repetitive work (yes vs no) | 1.3 | 0.6 to 2.2 |
| Godin (2005), Belgium, private and public sector, n=1521[18] | SHI: upper quartile | Effort-Reward Imbalance Questionnaire (highest quartile vs rest) | ♂ 3.4 ♀ 2.0 | 1.7 to 6.7 0.9 to 4.1 |
| Kivimaki (2007), Finland, government, n=21 271[19] | GHQ-12≥4 | Effort-Reward Imbalance Questionnaire (highest vs lowest quartile) | 2.04 | 1.80 to 2.32 |
| | | Procedural and organisational injustice: Organizational Justice Scale (highest vs lowest quartile) | 1.81 | 1.60 to 2.06 |
| | | Relational injustice: Organizational Justice Scale (highest vs lowest quartile) | 1.50 | 1.32 to 1.70 |
| Kivimaki (2007), Finland, hospital, n=10 736[19] | GHQ-12≥4 | Effort-Reward Imbalance Questionnaire (highest vs lowest quartile) | 1.59 | 1.24 to 2.05 |
| | | Procedural injustice: OJQ (highest vs lowest quartile) | 1.67 | 1.29 to 2.15 |
| | | Relational injustice: OJQ (highest vs lowest quartile) | 1.56 | 1.21 to 2.02 |

**Table 1** Continued

| Author, country population | Outcome | Exposure | | |
|---|---|---|---|---|
| | Definition and assessment | Definition and assessment | OR | (95% CI) |
| Hanson (2008), Sweden, various workers, n=3004[2] | MBI-GS≥75th percentile | Job demands: SWES | ♂ 2.09 | 1.52 to 2.88 |
| | | (≥2 positive vs <2 out of four items) | ♀ 1.79 | 1.36 to 2.35 |
| | | Decision authority: SWES | ♂ 1.36 | 0.98 to 1.88 |
| | | (≥2 positive vs <2 out of four items) | ♀ 1.41 | 1.07 to 1.86 |
| | | Coworker support: SWES; one item | ♂ 1.45 | 0.97 to 2.17 |
| | | (4-point Likert scale, dichotomised: low (1,2) vs high(3.4)) | ♀ 1.92 | 1.25 to 2.93 |
| | | Supervisor support: SWES; one item | ♂ 1.65 | 1.19 to 2.31 |
| | | (4-point Likert scale dichotomised: low (1,2) vs high(3.4)) | ♀ 1.22 | 0.91 to 1.65 |
| Devereux (2011), England, 11 industrial types, n=1463[20] | GHQ-12 GHQ-12>3 | Job demands questionnaire: 4 items (highest vs lowest tertile) | RR 1.62 | 1.26 to 2.09 |
| | | Decision latitude: 15 items (lowest vs highest tertile) | RR 1.11 | 0.86 to 1.42 |
| | | Coworker and supervisor support: 7 items (lowest vs highest tertile) | RR 1.47 | 1.18 to 1.84 |
| Sundin (2011), Sweden, nurses, n=555[21] | MBI Swedish version, EE EE ≥27 | Job demands: SWES, one item, 5-point Likert scale (at least once a week or more vs ≤1 day out of 10) | 4.33 | 1.98 to 9.45 |
| | | Coworker support: SWES; one item, 4-point Likert scale (less (1,2,3) vs always receiving support (4)) | 2.21 | 0.88 to 5.56 |
| | | Supervisor support: SWES; one item, 4-point Likert scale (less (1,2,3) vs always receiving support (4)) | 2.17 | 0.65 to 7.26 |
| Inoue (2013), Japan, manufacturing, n=569[3] | Kessler Psychological Distress Scale (K6) ≥5 K6 scale | Procedural injustice: Organizational Justice Questionnaire (OJQ): 7 items, 5 point scale; Permanent (P) non-permanent worker (NP) (highest vs lowest tertile) Relational injustice: OJQ interactional justice 6-items, 5 point scale (highest vs lowest tertile) | ♂ 1.37 (P) ♀ 5.25 (P) ♀ 2.84 (NP) ♂ 2.39 (P) ♀ 2.35 (P) ♀ 1.61 (NP) | 0.43 to 4.34 0.82 to 33.6 1.19 to 6.75 0.61 to 9.39 0.51 to 10.9 0.78 to 3.30 |
| Laine (2014), Finland, public sector, n=3298[22] | GHQ-12 ≥3 GHQ-12 | Procedural injustice: 4 items Moorman's inventory (highest vs lowest quartile) | 1.65 | 1.21 to 2.24 |
| | | Relational injustice: 4 items Moorman's inventory(highest vs lowest quartile) | 1.29 | 0.97 to 1.72 |
| | | Bullying: item about mental violence or workplace bullying (yes vs no) | 1.57 | 0.97 to 2.53 |
| Taniguchi (2015), Japan, elderly facilities, n=543[23] | BJSQ>13 (♂) or >12 (♀) | Bullying: Japanese NAQ, person-related | 3.46 (PR) | 1.49 to 8.05 |
| | | bullying (PR; six items), work-related bullying (WR; three items), | 2.85 (WR) | 0.61 to 13.26 |
| | | sexual harassment (SH; three items), five-point scale (≥1 item positive vs none) | 1.73 (SH) | 0.98 to 3.08 |
| Andersen (2017), Denmark, prison personnel, n=1741[24] | COPSOQ-CBI highest quartile | COPSOQ, 5-point Likert-scale (most exposed quartile vs least exposed three quartiles) | | |
| | | Job demands: 4 items | 1.61 | 1.21 to 2.16 |
| | | Coworker and supervisor support: 6 items | 1.31 | 0.98 to 1.76 |
| | | Emotional demands: 4 items | 1.46 | 1.06 to 2.01 |
| | | Effort-reward imbalance (low recognition): 3 items | 1.33 | 0.98 to 1.80 |

Continued

| Author, country population | Outcome | Exposure | | |
|---|---|---|---|---|
| | Definition and assessment | Definition and assessment | OR | (95% CI) |
| Oshio (2017), Japan, 12 industrial types, n=7419[25] | Kessler Psychological Distress Scale (K6) K6 ≥5 | Job demands: 5 items Japanese JCQ, 4- | ♂ 1.62 | 1.45 to 1.81 |
| | | point scale (*median-split method classifying high vs low*) | ♀ 1.71 | 1.38 to 2.13 |
| | | Procedural injustice: Japanese version of | ♂ 1.68 | 1.51 to 1.87 |
| | | OJQ, 7 items, 5-point Likert scale | ♀ 1.80 | 1.45 to 2.23 |
| | | Relational injustice: Japanese version of | ♂ 1.58 | 1.42 to 1.77 |
| | | OJQ, 6 items, 5-point Likert scale | ♀ 1.71 | 1.37 to 2.13 |
| | | Effort reward imbalance: Japanese version Effort-Reward Imbalance | ♂ 2.05 | 1.84 to 2.29 |
| | | Questionnaire, effort (three items), reward (seven items), 4-point scale | ♀ 1.84 | 1.48 to 2.28 |
| Kind (2018), Switzerland, youth welfare, n=121[26] | BOSS T-score ≥60 | Workplace aggression: exposure to verbal and physical threats *Verbal aggression vs no aggression* | HR 1.67 | 1.09 to 2.58 |
| Pihl-Thingvad (2019), Denmark, social educators, n=1823[27] | CBI≥75th percentile | Workplace violence: Scandinavian checklist, 4 items, 4-point Likert-scale (*high exposed (≥1 incident per month) vs non- exposed*) | 1.4 | 0.9 to 2.3 |

BJSQ, Brief Job Stress Questionnaire; BOSS, Burnout screening scales; CBI, Copenhagen Burnout Inventory; COPSOQ, Copenhagen Psychosocial Questionnaire; EE, Emotional Exhaustion; GHQ, General Health Questionnaire; JCQ, Job Content Questionnaire; MBI, Maslach Burnout Inventory; MBI-GS, Maslach Burnout Inventory-General Survey; NAQ, Negative Acts Questionnaire; OJQ, Organizational Justice Questionnaire ; QPJW, Questionnaires; Perception and Judgement of Work; RR, relative risk; SHI, Short Fatigue Inventory; SSPI, Setterlind Stress Profile Inventory; SWES, Swedish Work Environment Survey.

### Meta-analyses and assessment of evidence

In total, 12 psychosocial risk factors from 14 studies[1–3 15–22 24 25] from a population of 73 874 workers from Belgium, Denmark, England, Finland, Japan, Netherlands and Sweden were used in the meta-analysis. Reported distress prevalence varied from 4.9%[17] to 45.8%.[3] The follow-up time varied from 1 to 5 years. Table 3 summarises the assessment of evidence concerning the psychosocial risk factors for stress-related disorders.

### *Effort-reward balance*

Six cohort studies[16 18 19 24 25] demonstrate that there is moderate-quality evidence that effort-reward imbalance increases the incidence of SRD, with a pooled OR of 1.91 (95% CI 1.70 to 2.15) (see table 3 and figure 2). No statistically significant difference between subgroups of men and women was found.[16 18 25]

### *Procedural and relational justice*

Five cohort studies[3 19 22 25] demonstrate that there is moderate-quality evidence that low procedural justice with a pooled OR of 1.73 (95% CI 1.61 to 1.86) and low relational justice with a pooled OR of 1.55 (95%CI 1.44 to 1.67) increase the incidence of SRD (see table 3 and figure 3). No statistically significant difference between subgroups of men and women was found.[3 25]

### *Job demands*

Eight cohort studies[1 2 15 16 20 21 24 25] demonstrate that there is moderate-quality evidence that high job demands increase the incidence of SRD, with a pooled OR of 1.55 (95% CI 1.41 to 1.71) (see table 3 and figure 4). No

statistically significant difference between subgroups of men and women was found.[1 2 15 16 25]

### *Coworker and supervisor support*

In total, five cohort studies demonstrate that there is moderate-quality evidence that low coworkers support[1 2 16 21] with a pooled OR 1.29 (95% CI 1.17 to 1.43) and low supervisor support[1 2 15 16 21] with a pooled OR 1.27 (95% CI 1.14 to 1.40) increase the incidence of SRD. No statistically significant difference between subgroups of men and women was found.[1 2 15 16] The combination of low coworker and supervisor support[20 24] resulted in a pooled OR 1.41 (95% CI 1.18 to 1.69) (see table 3).

### *Emotional demands*

Two cohort studies[1 24] demonstrate that there is moderate-quality evidence that high emotional demands increase the incidence of SRD, with a pooled OR of 1.58 (95% CI 1.35 to 1.84) (see table 3). No statistically significant difference between men and women was found in the study of Bültmann *et al*.[1]

### *Decision authority*

Two cohort studies[2 16] demonstrate that there is moderate-quality evidence that low decision autonomy increases the incidence of SRD with a pooled OR of 1.34 (95% CI 1.20 to 1.49) (see table 3). No statistically significant difference between subgroups of men and women was found.

### *Job security, decision latitude, skill discretion and bullying*

One cohort study[1] demonstrates that there is low-quality evidence that job insecurity increases the incidence of

**Table 2** Methodological quality scores of 9 items for studies regarding risk factors*

| Author | 1 | 2 | 3 | 4 | 5 | 6 | 7 | 8 | 9 | Summary score |
|---|---|---|---|---|---|---|---|---|---|---|
| Mino (1999)[15] | – | + | – | – | + | + | + | –† | + | 5 |
| Stansfeld (1999)[16] | – | + | + | – | + | + | + | –‡ | + | 6 |
| Bültmann (2002)[1] | + | – | + | + | – | + | + | –§ | + | 6 |
| Bonde (2005)[17] | + | + | + | – | + | + | + | +¶ | + | 8 |
| Godin (2005)[18] | + | – | + | + | – | + | + | –** | + | 6 |
| Kivimaki 10-town study (2007)[19] | + | + | + | – | – | + | + | –†† | + | 6 |
| Kivimaki hospital study (2007)[19] | + | + | + | + | – | + | + | –†† | + | 7 |
| Magnusson Hanson (2008)[2] | + | + | – | – | – | + | + | –‡‡ | + | 5 |
| Devereux (2011)[20] | + | ? | – | + | – | + | + | –§§ | + | 5 |
| Sundin (2011)[21] | + | + | – | + | – | + | + | +¶¶ | + | 7 |
| Inoue (2013)[3] | + | + | + | + | – | + | + | +*** | + | 8 |
| Laine (2014)[22] | – | + | + | + | – | + | + | +††† | + | 7 |
| Taniguchi (2015)[23] | + | + | – | + | – | + | + | +‡‡‡ | + | 7 |
| Andersen (2017)[24] | + | + | – | + | – | + | + | +§§§ | + | 7 |
| Oshio (2017)[25] | + | + | + | + | – | + | + | +¶¶¶ | + | 8 |
| Kind (2018)[26] | + | – | + | + | + | + | + | +**** | + | 8 |
| Pihl-Thingvad (2019)[27] | + | + | + | + | – | + | + | +†††† | + | 8 |
| Total item score | 14 | 13 | 11 | 12 | 4 | 17 | 17 | 9 | 17 | |

*Criteria and scoring options for quality score for these nine items are reported in the Methods section.
†Adjusted for age, sex, family life satisfaction, perceived physical health.
‡Adjusted for age, sex, employment grade and baseline GHQ score.
§Adjusted for age, sex, education, living alone, employment status, presence of disease, baseline fatigue score.
¶Adjusted for age, sex, body mass index, leisure time activity, pain threshold, marital status, psychiatric disorder.
**Adjusted for age, sex, education, threat from global economy, job dissatisfaction, workplace instability.
††Adjusted for age, sex, occupational status.
‡‡Adjusted for age, sex, marital status, country of birth, social class, physical exhaustion.
§§Adjusted for age, sex, shift work.
¶¶Adjusted for age, sex, marital status, years of (current) employment (only for job demands).
***Adjusted for age, sex, education, marital status, chronic physical diseases, occupation, life events, neuroticism.
†††Adjusted for age, sex, socioeconomic position, marital status, employment, health behaviour, limiting longstanding illness, physical work.
‡‡‡Adjusted for age, sex, job carrier, occupation, marital status, employment, work shift, smoking status.
§§§Adjusted for age, sex, marital status, employment years, occupational characteristics and exposures.
¶¶¶Adjusted for age, sex, education, occupation, hours worked per week, household income, family member to share living expenses, firm codes.
****Adjusted for age, sex, work experience, employment years, private stressors.
††††Adjusted for age, sex, somatic and mental health at baseline, lifestyle factors, work-related factors.
GHQ, General Health Questionnaire.

SRD, with a pooled OR of 1.63 (95% CI 1.18 to 2.27) for male and no increase of the incidence for female with a pooled OR of 0.94 (95% CI 0.56 to 1.59).

Two cohort studies[1 20] demonstrate that there is moderate-quality evidence for no increased incidence of SRD due to low decision latitude, with an OR of 1.07 (95% CI 0.92 to 1.25). Two cohort studies[16 17] demonstrate that there is moderate-quality evidence for no increased incidence of SRD due to low skill discretion, with an OR of 1.11 (95% CI 0.94 to 1.32). Four cohort studies[22 23 26 27] demonstrate very low-quality evidence that there is inconsistent evidence for increased incidence of SRD with different types of bullying and workplace violence with ORs varying from 1.42 (95% CI 0.42 to 4.79) to 3.64 (95% CI 0.83 to 15.92) (see table 3). Laine et al[22] studied workplace bullying or mental violence as the isolation of a member of the organisation, the underestimation of work performance, being threatened, being talked about behind one's back or other forms of pressure. Taniguchi et al[23] studied person-related bullying, workplace bullying and sexual harassment. Kind et al[26] studied verbal and physical client aggression and Phil-Thingvad et al[27] studied physical violence.

## DISCUSSION
### Main findings
This systematic review, including a meta-analysis of prospective cohort studies revealed moderate-quality evidence that effort-reward imbalance, low procedural and relational justice, high job demands, low coworker and supervisor support, high emotional demands and low decision authority increase the incidence of stress-related disorders, varying from 20% to 90%. Low-quality evidence was found for an association between job insecurity and stress-related disorder among men. Moderate

**Table 3** Quality of the evidence for the relationship between risk factors and stress related disorders according to the GRADE framework

| | Number of participants | Number of studies/ cohorts | Study phase* | Study limitations Study quality majority of studies <11/16: ↓ | Inconsistency I² ≥50%: ↓ | Indirectness Yes: ↓ | Imprecision CI effect size (<1 and>2) Yes: ↓ | Publication Bias Yes: ↓ | Effect size OR (95% CI) Lower Limit OR >2.0: ↑ | Exposure-response gradient (dose-effect) majority of studies: ↑ | Overall quality of evidence |
|---|---|---|---|---|---|---|---|---|---|---|---|
| Effort-reward imbalance | 76 760 | 6/6 | 1 | No | 48% | No | No | No | 1.91 (1.70 to 2.15) | 2/6 | Moderate |
| Low procedural justice | 64 676 | 5/5 | 1 | No | 0% | No | No | No | 1.74 (1.62 to 1.86) | 2/5 | Moderate |
| Low relational justice | 64 676 | 5/5 | 1 | No | 0% | No | No | No | 1.55 (1.44 to 1.67) | 2/5 | Moderate |
| High job demands | 41 397 | 8/8 | 1 | No | 49% | No | No | No | 1.56 (1.41 to 1.72) | 0/8 | Moderate |
| Low coworker support | 22 920 | 4/4 | 1 | No | 10% | No | No | No | 1.29 (1.17 to 1.43) | 0/4 | Moderate |
| Low supervisor support | 23 382 | 5/5 | 1 | No | 0% | No | No | No | 1.27 (1.16 to 1.38) | 0/5 | Moderate |
| Low coworker and supervisor support | 7262 | 2/2 | 1 | No | 0% | No | No | No | 1.41 (1.18 to 1.69) | 0/2 | Moderate |
| High emotional demands | 13 641 | 2/2 | 1 | No | 0% | No | No | Undetected | 1.58 (1.35 to 1.84) | 0/2 | Moderate |
| Low decision authority | 13 312 | 2/2 | 1 | No | 0% | No | No | Undetected | 1.34 (1.20 to 1.49) | 0/2 | Moderate |
| Job insecurity | 8833 | 1/1 | 1 | No | – | No | Yes: ↓ | Undetected | Men 1.63 (1.18 to 2.27) Women 0.94 (0.56 to 1.59) | 0/1 | Low |
| Decision latitude | 11 287 | 2/2 | 1 | No | 0% | No | No | Undetected | 1.07 (0.92 to 1.25) | 0/2 | Moderate |
| Low skill discretion | 3123 | 2/2 | 1 | No | 0% | No | No | Undetected | 1.11 (0.94 to 1.32) | 0/2 | Moderate |
| Bullying and violence | 9694 | 4/4 | 1 | No | Heterogeneity in definitions | No | Yes: ↓ | No | 1.42–3.64 (0.42 to 15.92) | 0/4 | Low |

*All included studies are phase one explanatory prospective cohort studies with 'moderate' as starting point for the quality of the evidence.

van der Molen HF, *et al. BMJ Open* 2020;**10**:e034849. doi:10.1136/bmjopen-2019-034849

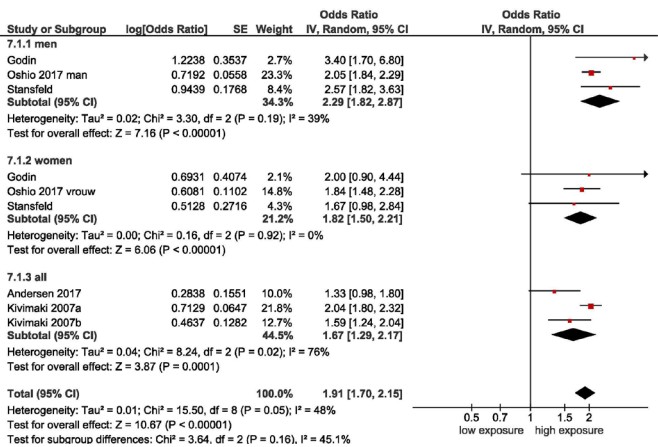

**Figure 2** Forest plot of studies regarding effort-reward imbalance and SRD. SRD, stress-related mental disorders.

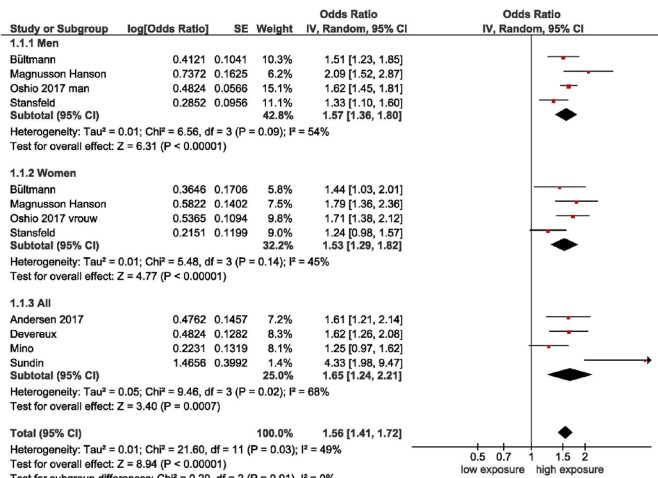

**Figure 4** Forest plot of studies regarding job demands and SRD. SRD, stress-related mental disorders.

to low-quality evidence suggesting no associations were found for decision latitude, skill discretion and bullying.

## Psychosocial risk factors and SRD

All prospective cohort studies were controlled for the personal factors of age and sex and non-work-related factors such as education and marital status. Six studies also controlled for other work-related psychosocial risk factors in various models.[3 19 20 22 27] Regarding these studies, we decided not to take into account the fully adjusted model in the meta-analyses in order to avoid overcorrection and strengthen transparency and comparability between studies. However, these models yielded small and non-statistically significant differences between the risk estimates.

All studies defined SRD based on self-report with validated and standardised questionnaires, that is, (shortened) GHQ,[1 15 16 19 20 22] Kessler 6,[3 25] Maslach Burnout Inventory,[2 21] Copenhagen Psychosocial Questionnaire-Copenhagen Burnout Inventory,[24 27] Setterlind Stress Inventory Profile,[17] fatigue inventory,[18] Brief Job Stress Questionnaire[27] and burnout screening scale.[26]

The pooled ORs of the meta-analyses of psychosocial risk factors were equal or slightly raised compared with the prior results of Nieuwenhuijsen et al[10] and in line with the findings of a recent meta-review of work-related risk

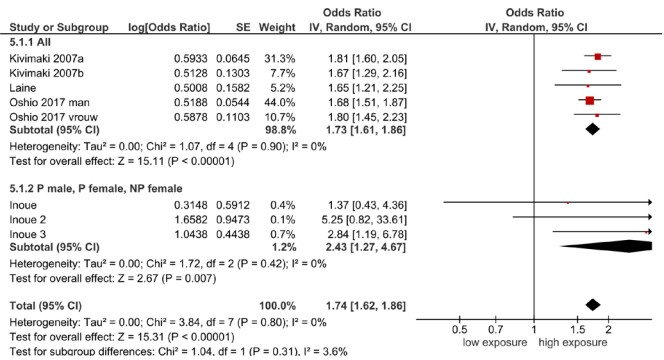

**Figure 3** Forest plot of studies regarding procedural injustice and SRD. SRD, stress-related mental disorders.

factors for common mental-health problems including depression and anxiety.[11]

## Methodological considerations

The exposure categories for the risk factors and the case definitions of SRD varied in the included studies. We have explicitly described all definitions and assessments of self-reported exposures and outcomes (see table 1). In general, psychological distress can stem from private life as well as from work. Therefore, we only included high-quality prospective studies aimed at identifying work-related risk factors, also controlling for other factors outside the work context. Although there is debate about the definition of burnout,[5 7 28] we regarded distress and burnout as outcomes of the same continuum. We found that similar work-related psychosocial risk factors were associated with distress and the one-dimensional measures of burn-out ('exhaustion'). Should more studies using the multiple dimension definition of burnout become available, then subgroup analyses of these studies can be performed to assess whether the work-related risk factors are similar in this diagnostic subgroup.

We used the risk estimates—mostly ORs—as reported in the original prospective cohort studies (see also table 1). Meta-analyses were performed to estimate pooled associations, although the variety in measurement instruments and dichotomisation rules introduced heterogeneity in the outcome measures (case definitions) and exposure categories of psychosocial risk factors. However, the exposure categories for the psychosocial risk factors included in the meta-analyses were already established in theoretical models of effort-reward imbalance,[29] job demands-control[30] and organisational justice.[31] Furthermore, the case definitions of SRD used validated questionnaires and the methodological quality of the included studies was high and accompanied by a low to moderate statistical heterogeneity across the pooled studies ($I^2$ varying from 0% to 49%) which justify the meta-analyses. With the low number of studies for each risk factor, we feel that more subgroup analyses—besides the meta-analyses for separate

risk factors and sexes—are not informative. Reporting bias is a concern because both psychosocial factors and SRD are self-reported in the included studies; it is conceivable that certain psychological states at baseline are associated both with the over-reporting of adverse working conditions at baseline and with the risk of onset of stress related disorders at follow-up.[32] In prospective studies, this risk is less pronounced;[33] however, in only 4 out 17 prospective studies in this systematic review, data on outcome were collected at least at three different time points.

The GRADE framework made it possible to assess to the quality of evidence in reviews of prognostic factor research. However, the starting level for grading of the evidence for prospective aetiological studies is subject to scientific debate. For intervention studies, the grading of observational studies starts with low evidence. For prospective observational studies, which can answer prognostic, or as in our systematic review, aetiological research questions, Huguet *et al*[14] suggest that higher levels of evidence are derived from cohort studies that seeks to confirm independent associations between the prognostic factor and the outcome (high level of evidence) or—as in our systematic review—aimed to identify associations between potential prognostic factors and the outcome (moderate level of evidence). Also, recent GRADE guidelines state that best evidence regarding prognostic factors usually originates from observational studies (eg, cohort studies or registries), while randomised control trials could include restrictions that exclude subjects relevant for the assessment of prognostic factors.[34]

Based on Huguet *et al*,[14] we classified the risk of publication bias as 'no bias' when the value of the risk factor in predicting the outcome has been repetitively investigated in explanatory studies. We provided no funnel plots because the number of cohort studies for each psychosocial risk factor was substantially lower than 10. Visually checking the funnel plots, however, suggested no serious publication bias or could not assessed (undetected publication bias).

For the risk factor, bullying and violence, it was decided not to perform a meta-analysis because of the different concepts of the psychosocial risk factor, that is, person-related bullying, work-related bullying, sexual harassment, workplace bullying, threatening or violence.

Eventually, further research in harmonising the assessment of SRD in cohort studies[7] and the clinical assessment with signs and symptoms of SRD,[35] combined with more detailed or objective task-based exposure assessment[36] will contribute to a better insight into the work-related psychosocial risk factors and their mediating or moderating factors.

## Prevention

To manage and prevent stress disorders, it is necessary to identify which psychosocial risk factors at work contribute to the onset of SRD. West *et al*[37] describe the importance of organisational strategies in their review on interventions to prevent and reduce SRD among physicians. Kalani *et al*[38] report in their meta-review that various studies of the effectiveness of individually and organisationally directed

interventions have led to different results on reducing physician burnout. Ultimately, to develop individually and organisationally directed interventions, it is necessary that the relevant psychosocial risk factors in organisations are known.[37 38]

Recently, Fan *et al*[39] have suggested that some psychosocial factors such as job control, job security and social support are also associated with a greater likelihood of workers' experiencing positive mental well-being in terms of satisfaction and purpose in life, personal growth, social contribution and integration. This study implicates the double value of workplace policies and practices that improve psychosocial working conditions, reduce work-related SRD and improve mental well-being in general, for example, by giving workers greater job control or social support.[39]

In conclusion, several psychosocial work-related risk factors for SRD were established, confirming the multifactorial aetiology of SRD. Effort-reward imbalance, low organisational justice and high job demands were associated with the largest increased risk of SRD, varying from 60% to 90%. Awareness of these risk factors could be the starting point for the selection of preventive interventions to reduce work-related SRD.

**Acknowledgements** We are very grateful to Joost Daams for his contribution to the literature search and Nicolaas Kylstra and Kim van der Molen for their help with screening activities.

**Contributors** HFvdM and GdG initiated the study, performed data extraction and statistical analyses and drafted the manuscript. All authors performed the study selection and interpretation of the data. All authors made substantial contributions to the conception of the study and manuscript.

**Funding** The work was sponsored by the Ministry of Social Affairs and Employment, the Netherlands. Grant number: 5100–24108

**Competing interests** None declared.

**Patient consent for publication** Not required.

**Provenance and peer review** Not commissioned; externally peer reviewed.

**Data availability statement** All data relevant to the study are included in the article or uploaded as supplementary information

**ORCID iD**
Henk F van der Molen http://orcid.org/0000-0002-0719-2020

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
