## [Reviewer comments · BMJ Open]

ARTICLE DETAILS

TITLE (PROVISIONAL)	Work-related psychosocial risk factors for stress-related mental disorders – an updated systematic review and meta-analysis
AUTHORS	van der Molen, Henk; Nieuwenhuijsen, Karen; Frings-Dresen, Monique; de Groene, Gerda

VERSION 1 - REVIEW

REVIEWER	Reiner Rugulies National Research Centre for the Working Environment, Copenhagen, Denmark
REVIEW RETURNED	30-Oct-2019

GENERAL COMMENTS	It is a little special to review this paper. As I understand the paper was first submitted to Occupational and Environmental Medicine, was rejected there after review, then the authors did a point-by-point response to the reviewers' concerns and then submitted this paper to BMJ Open. So, this is now a kind of mixed paper, it is an initial submission with regard to BMJ Open, but is also a revised submission, because the authors took the reviewers comments to the previous OEM submission into consideration. In evaluating the paper, it needs also to be considered that BMJ Open has different standards for accepting manuscripts for publication than OEM. In my review, I will take into consideration the authors' response to the three previous reviews. 1) The authors admit in their response letter that this review was indeed an update of a previous review, as one of the reviewers had suspected and they now make this more clear in the manuscript. However, this is not enough. It should be made clear both in the title of the paper and in the abstract that this is an update. Further, it should be made clear in the title that this manuscript is not about all work-related psychosocial risk factors but is about selected risk factors. Thus, the title should be re-phrased like something like this: "Selected work-related psychosocial risk factors for stress-related disorders – an updated systematic review and meta-analysis". In the abstract it should say under objectives: "The objective is to conduct an update of a previously published review and meta-analysis on the association between selected work-related psychosocial risk factors and risk of stress-related disorders (SRD). 2) Also in the introduction, it needs to be made clear that there was this previous review by Nieuwenhuijsen et al. 2010, the main results of this previous review should be reported and a rationale should be given, why an update of this review is warranted.
---

3) I agree with Reviewer #2 comment #6: Publication bias is an issue here and the authors need to address this. Usually, funnel plots for visually inspecting publication bias would be added to the manuscript. Assessing publication bias is also required by PRISMA guidelines. As I understand, the authors did not do this, because it is often recommended that funnel plots are based on at least 10 studies, a requirement that was not met here. Instead, the authors downgraded all factors because of publication bias. While I appreciate that the authors want to be conservative here, this general downgrading for unclear publication is not satisfactory. I encourage the authors to find an alternative here and to assess publication bias for each exposure-outcome relation and not general for all exposure-outcome relations.

4) I suggest a summarizing section in the result section. To write, for example: "To summarize, we found moderate evidence for an association with risk of stress-related disorders for the following psychosocial work factors: a, b, c. We found low evidence for the work factors d, e, f."

5) I agree with Reviewer #1, comment #7 that it is highly unusual to start with "high level of evidence" for observational studies. The authors write in their response to the reviewer "Indeed for intervention studies the starting point for observation studies would be low", a sentence that I find difficult to comprehend. The usual procedure for GRADE is to start with low evidence for observational studies, see for example the review by Theorell et al. 2015 on work environment and depression (<https://www.ncbi.nlm.nih.gov/pubmed/26232123>). If the authors want to claim that they are using GRADE they should adhere to what is common practice in GRADE. If they authors want to use a different grading system that starts with "high level of evidence" for observational studies, then they should not call this GRADE. Otherwise, this would be misleading for readers. I find it problematic to start with high level of evidence for observational studies, there is too much concern about substantial bias due to unmeasured confounding. Just think about the story about hormone replacement therapy and risk of cardiovascular disease. And the risk of bias due to unmeasured confounding is even higher here when it comes to working conditions and stress-related disorders than it was in the hormone replacement therapy studies, I believe. So, I strongly advise against starting quality grading with "high level of evidence" here.

6) Finally, the discussion section should more comprehensively discuss possible biases in the studies. In their response to the reviewers, the authors cite articles by Rugulies et al. 2017, Scandinavian Journal of Work, Environment and Health <https://www.ncbi.nlm.nih.gov/pubmed/28306759> and Bonde 2008, OEM <https://www.ncbi.nlm.nih.gov/pubmed/18417557>. If the authors read these two reviews carefully, they will see that these review papers discuss at length possible biases in the studies they had reviewed, in particular with regard to reporting bias. And since in the studies in this review here, both exposure and endpoint are self-reported and since it is conceivable that certain psychological states (e.g., low mood) at baseline is associated both with the overreporting of adverse working conditions at baseline and with the risk of onset of stress-disorders at follow-up, reporting bias is a major concern. Thus, a critical discussion of the limitations of the reviewed papers, in particular with regard to biases that may have inflated associations, is needed.

REVIEWER	Johannes Siegrist Centre for Health and Society, Faculty of Medicine Heinrich-Heine-University Düsseldorf Germany
REVIEW RETURNED	02-Dec-2019

GENERAL COMMENTS	This manuscript is a re-submission of an earlier version that has now been substantially improved in response to Reviewers' comments. One major previous concern was the definition of outcomes of this review. This concern has now been largely resolved by the authors' elaborations in the text as well as in their Response to Reviewer comments. However, the term 'stress-related disorders' (SRD) would need a further precision as it seems to be restricted to mental ill health. This should be explicitly stated in view of large evidence indicating that some somatic disorders (such as coronary heart disease) are often also labelled stress-related disorders. Another important comment concerns the grading of quality of evidence according to GRADE. One Reviewer rightly stated that observational studies cannot be graded as 'high evidence'. I fully agree with this Reviewer and propose that authors revise their judgment from 'high' to 'moderate'. Even observational prospective studies do not allow a higher grading as evidence on causality is restricted. See e.g. Lam J et al. (2014) The Navigation Guide. Environ Health Perspect 122: 1040-51; Schünemann H et al. (2011) The GRADE Approach and Bradford Hill criteria for causality. J Epidemiol Community Health 65: 312-15. Apart from these critical remarks the manuscript in its current Version is now ready for publication according to my evaluation.
--

REVIEWER	Chris Peterson La Trobe University Australia
REVIEW RETURNED	04-Feb-2020

GENERAL COMMENTS	This study reports on a systematic review and meta-analysis examining the work psychosocial risk factors related to stress – associated disorders. This was a well presented paper. Line 201 'Our starting point for the quality of the evidence was 'high' for longitudinal studies that sought to confirm independent associations between the prognostic factor and the outcome ('Phase 2' explanatory studies)'. Comment: The rating of longitudinal studies as high is appropriate. Line 335 'Although there is debate about the definition of burnout [5,7, 28], we regarded distress and burnout as outcomes of the same continuum. We found that similar work-related psychosocial risk factors were associated with distress and the one-dimensional measures of burn-out ('exhaustion')'. Comment: I would debate addressing stress and burnout as similar/the same. They can have different symptoms
---

	Line 340 'We used the risk estimates as reported in the original longitudinal studies (see also table 1; i.e. OR, RR and HR). Mostly they reported ORs, possibly because of the mathematical advantages of using OR' Comment: The authors have appropriately discussed using Odds Ratios from longitudinal studies where Risk Ratios have not been provided. In addition there was an appropriate defence of using two studies as a minimum in the meta – analysis. Line 358 'For the risk factor, bullying and violence, it was decided not to perform a meta-analysis because of the different concepts of the psychosocial risk factor, i.e. person-related bullying, work-related bullying, sexual harassment, workplace bullying, threatening or violence'. Comment: Given the concerns of the authors it would be useful to have an analysis of bullying, given that there are many types of bullying as they may share the same symptoms.
--	---

REVIEWER	Christiana Kartsonaki University of Oxford
REVIEW RETURNED	13-Feb-2020

GENERAL COMMENTS	Statistical review of "Work-related psychosocial risk factors for stress-related disorders – a systematic review and meta-analysis" The statistical analysis appears appropriate but there are a few issues to clarify:  1. Lines 182-186: How was heterogeneity assessed? 2. Lines 194-197: Did all included studies report ORs? 3. I suggest assessing publication bias. 4. Lines 340-342: If different measures of association were used, this should have been described in the methods (see comment 2). Are the included studies all prospective, i.e. they record exposures and follow individuals up for a period of time until outcomes are observed? Or do some have different study designs? It is not clear from the paper. They are also referred to as longitudinal. Also it is not clear what the mathematical advantages of using ORs are.
---

VERSION 1 – AUTHOR RESPONSE

Reviewer #1

As I understand the paper was first submitted to Occupational and Environmental Medicine, was rejected there after review, then the authors did a point-by-point response to the reviewers' concerns and then submitted this paper to BMJ Open. So, this is now a kind of mixed paper, it is an initial submission with regard to BMJ Open, but is also a revised submission, because the authors took the reviewers comments to the previous OEM submission into consideration. In evaluating the paper, it needs also to be considered that BMJ Open has different standards for accepting manuscripts for

publication than OEM. In my review, I will take into consideration the authors' response to the three previous reviews.

► Thanks you for the time of reviewing. It was indeed a request of BMJ Open also to take into account the OEM reviews after the suggestion of OEM to directly forward the manuscript to BMJ open.

1) The authors admit in their response letter that this review was indeed an update of a previous review, as one of the reviewers had suspected and they now make this more clear in the manuscript. However, this is not enough. It should be made clear both in the title of the paper and in the abstract that this is an update. Further, it should be made clear in the title that this manuscript is not about all work-related psychosocial risk factors but is about selected risk factors. Thus, the title should be re-phrased like something like this: "Selected work-related psychosocial risk factors for stress-related disorders – an updated systematic review and meta-analysis". In the abstract it should say under objectives: "The objective is to conduct an update of a previously published review and meta-analysis on the association between selected work-related psychosocial risk factors and risk of stress-related disorders (SRD).

► We adapted the title and objective with the suggested wording of 'update'. We prefer not to add the wording of 'selective' psychosocial risk factors, since all psychosocial risk factors we eligible for inclusion (see lines 120-125), as was the case in the earlier review of Nieuwenhuijsen et al., and indeed we found also other risk factors, e.g. bullying, which were not found in the earlier review.

2) Also in the introduction, it needs to be made clear that there was this previous review by Nieuwenhuijsen et al. 2010, the main results of this previous review should be reported and a rationale should be given, why an update of this review is warranted.

► We added the results of the previous review (see lines 96-102); a rationale was already given (see lines 93-96).

3) I agree with Reviewer #2 comment #6: Publication bias is an issue here and the authors need to address this. Usually, funnel plots for visually inspecting publication bias would be added to the manuscript. Assessing publication bias is also required by PRISMA guidelines. As I understand, the authors did not do this, because it is often recommended that funnel plots are based on at least 10 studies, a requirement that was not met here. Instead, the authors downgraded all factors because of publication bias. While I appreciate that the authors want to be conservative here, this general downgrading for unclear publication is not satisfactory. I encourage the authors to find an alternative here and to assess publication bias for each exposure-outcome relation and not general for all exposure-outcome relations.

► We indeed not provided funnel plots because of the low number of studies. However, we agree with the reviewer that downgrading for unclear publication bias is too general. Following Huguet et al. (2013) we defined no publication bias when the value of the risk factor in predicting the outcome has been repetitively investigated in explanatory studies. We also checked visually the funnel plots and found no serious indications of publication bias or could not assess this because of the very low number of included studies [undetected publication bias] for some psychosocial factors. (see lines 370-374 and table 3).

4) I suggest a summarizing section in the result section. To write, for example: “To summarize, we found moderate evidence for an association with risk of stress-related disorders for the following psychosocial work factors: a, b, c. We found low evidence for the work factors d, e, f.”

► Thank you for the suggestion, however we prefer the present text with the summarizing table 3 and the start of the discussion with the main findings as the reviewer suggested. If the editor wishes so we can replace this paragraph to the end of the result section.

5) I agree with Reviewer #1, comment #7 that it is highly unusual to start with “high level of evidence” for observational studies. The authors write in their response to the reviewer “Indeed for intervention studies the starting point for observation studies would be low”, a sentence that I find difficult to comprehend. The usual procedure for GRADE is to start with low evidence for observational studies, see for example the review by Theorell et al. 2015 on work environment and depression (<https://eur04.safelinks.protection.outlook.com/?url=https%3A%2F%2Fwww.ncbi.nlm.nih.gov%2Fpubmed%2F26232123&data=02%7C01%7Ch.f.vandermolen%40amsterdammc.nl%7C4b1f9944e2734eddb99408d7b4912bfd%7C68dfab1a11bb4cc6beb528d756984fb6%7C0%7C0%7C637176407912335031&sdata=6A1jCawxGLkzG8GVRAsndZv2VAvq11VFthGwBG4d5dw%3D&reserved=0>). If the authors want to claim that they are using GRADE they should adhere to what is common practice in GRADE. If they authors want to use a different grading system that starts with “high level of evidence” for observational studies, then they should not call this GRADE. Otherwise, this would be misleading for readers. I find it problematic to start with high level of evidence for observational studies, there is too much concern about substantial bias due to unmeasured confounding. Just think about the story about hormone replacement therapy and risk of cardiovascular disease. And the risk of bias due to unmeasured confounding is even higher here when it comes to working conditions and stress-related disorders than it was in the hormone replacement therapy studies, I believe. So, I strongly advise against starting quality grading with “high level of evidence” here.

► Following Huguet et al. 2013 we originally started the evidence for prognostic studies as high for explanatory research aimed to confirm independent associations between potential prognostic factor and the outcome (phase 2 explanatory study). However due to concerns of several reviewers we checked again and decided to start at moderate because we aim to identify associations between potential prognostic factors (in our review risk factors) and the outcome (phase 1 explanatory study) while not all included studies aim to confirm independent associations between risk factors and outcome. So we changed the start of evidence into moderate in the method section (see lines 200-203 and table 3).

We also contacted professionals using GRADE and checked the literature. Unfortunately there is no consensus document for using GRADE for prognostic etiologic studies but there is indeed increasing support for the notion that the start for prognostic cohort studies should not be low as it would be in intervention studies (see Grade guideline 28 from Foroutana et al. 2020 in Journal of Clinical Epidemiology). Furthermore, this systematic review was aimed to identify associations and no prevention or intervention research questions were included. Also the provided reference of Theorell by the reviewer adapted the GRADE approach by upgrading the scientific evidence when there was strong coherence of results between studies according to Cochrane guidelines. All in all, we feel that starting the GRADE procedure with moderate best suits our purpose and is in line with current views on GRADE in this ongoing debate. We also discussed this lack of consensus in the discussion section (see lines 359-369).

6) Finally, the discussion section should more comprehensively discuss possible biases in the studies. In their response to the reviewers, the authors cite articles by Rugulies et al. 2017, Scandinavian Journal of Work, Environment and Health
<https://eur04.safelinks.protection.outlook.com/?url=https%3A%2F%2Fwww.ncbi.nlm.nih.gov%2Fpubmed%2F28306759&data=02%7C01%7Ch.f.vandermolen%40amsterdamumc.nl%7C4b1f9944e2734eddb99408d7b4912bfd%7C68dfab1a11bb4cc6beb528d756984fb6%7C0%7C0%7C637176407912335031&sdata=7uYKpr6De8K12zGUxhVB9PnBu8Oe4%2BN%2BSxPsYbNV96s%3D&reserved=0> and Bonde 2008, OEM
<https://eur04.safelinks.protection.outlook.com/?url=https%3A%2F%2Fwww.ncbi.nlm.nih.gov%2Fpubmed%2F18417557&data=02%7C01%7Ch.f.vandermolen%40amsterdamumc.nl%7C4b1f9944e2734eddb99408d7b4912bfd%7C68dfab1a11bb4cc6beb528d756984fb6%7C0%7C0%7C637176407912335031&sdata=TreRhYce7ljiZCRPLJxEGgk%2FSmbw9EzpXi9MAgNz6Ws%3D&reserved=0>. If the authors read these two reviews carefully, they will see that these review papers discuss at length possible biases in the studies they had reviewed, in particular with regard to reporting bias. And since in the studies in this review here, both exposure and endpoint are self-reported and since it is conceivable that certain psychological states (e.g., low mood) at baseline is associated both with the overreporting of adverse working conditions at baseline and with the risk of onset of stress-disorders at follow-up, reporting bias is a major concern. Thus, a critical discussion of the limitations of the reviewed papers, in particular with regard to biases that may have inflated associations, is needed.

► Indeed a bias is the self-report of determinants and outcomes. This is a timely remark and obliging the suggestion of the reviewer, we added the reporting bias of certain psychological states (e.g., low mood) at baseline could be associated both with the over reporting of adverse working conditions at baseline and with the risk of onset of stress-disorders at follow-up. However, in prospective studies this risk is less pronounced (see lines 353-357). Furthermore, we only included studies where the population did not have a certain psychological state at baseline, which limits the risk of a psychological state at baseline influencing the reporting of adverse working conditions at baseline.

We feel that the main limitations concerning lack of harmonized risk factors and outcomes are already sufficiently described. In addition, the Grade assessment already includes the assessment of different types of bias through the starting of moderate evidence taken into account the phase 1 explanatory studies (see lines 200-203).

Reviewer #2

This manuscript is a re-submission of an earlier version that has now been substantially improved in response to Reviewers' comments. One major previous concern was the definition of outcomes of this review. This concern has now been largely resolved by the authors' elaborations in the text as well as in their Response to Reviewer comments. However, the term 'stress-related disorders' (SRD) would need a further precision as it seems to be restricted to mental ill health. This should be explicitly stated in view of large evidence indicating that some somatic disorders (such as coronary heart disease) are often also labelled stress-related disorders.

► We clarified this in the title, abstract and introduction by adding 'mental' in terms of stress related mental disorders.

Another important comment concerns the grading of quality of evidence according to GRADE. One Reviewer rightly stated that observational studies cannot be graded as 'high evidence'. I fully agree

with this Reviewer and propose that authors revise their judgment from 'high' to 'moderate'. Even observational prospective studies do not allow a higher grading as evidence on causality is restricted. See e.g. Lam J et al. (2014) The Navigation Guide. Environ Health Perspect 122: 1040-51; Schünemann H et al. (2011) The GRADE Approach and Bradford Hill criteria for causality. J Epidemiol Community Health 65: 312-15.

► We acknowledge the provided references. However, these are referring to intervention studies or experimental studies where randomization is possible; we agree that the start grading should be low for observational intervention studies.

Following Huguet et al. 2013 we originally started the evidence for prognostic studies as high for explanatory research aimed to confirm independent associations between potential prognostic factor and the outcome (phase 2 explanatory study). However due to concerns of several reviewers we checked again and decided to start at moderate because we aim to identify associations between potential prognostic factors (in our review risk factors) and the outcome (phase 1 explanatory study) while not all included studies aim to confirm independent associations between risk factors and outcome. So we changed the start of evidence into moderate in the method section (see lines 200-203 and table 3).

We also contacted professionals using GRADE and checked the literature. Unfortunately there is no consensus document for using GRADE for prognostic etiologic studies but there is indeed increasing support for the notion that the start for prognostic cohort studies should not be low as it would be in intervention studies (see Grade guideline 28 from Foroutana et al. 2020 in Journal of Clinical Epidemiology). Furthermore, this systematic review was aimed to identify associations and no prevention or intervention research questions were included. Also the provided reference of Theorell by the reviewer adapted the GRADE approach by upgrading the scientific evidence when there was strong coherence of results between studies according to Cochrane guidelines. All in all, we feel that starting the GRADE procedure with moderate best suits our purpose and is in line with current views on GRADE in this ongoing debate. We also discussed this lack of consensus in the discussion section (see lines 359-369).

Apart from these critical remarks the manuscript in its current version is now ready for publication according to my evaluation.

► Thank you.

Reviewer #3

This was a well presented paper.

► Thank you.

Line 201 'Our starting point for the quality of the evidence was 'high' for longitudinal studies that sought to confirm independent associations between the prognostic factor and the outcome ('Phase 2' explanatory studies)'.
'

Comment: The rating of longitudinal studies as high is appropriate.

► Thank you. However due to concerns of several reviewers we again checked and decided to start at 'moderate' because we aimed to identify associations between potential prognostic factors (in our review risk factors) and the outcome (phase 1 explanatory study) and not all included studies aimed to confirm independent associations between risk factors and outcome (see lines 200-203). We also discussed the lack of consensus in the grading of evidence in the discussion section (see lines 359-369).

Line 335 'Although there is debate about the definition of burnout [5,7, 28], we regarded distress and burnout as outcomes of the same continuum. We found that similar work-related psychosocial risk factors were associated with distress and the one-dimensional measures of burn-out ('exhaustion').

Comment: I would debate addressing stress and burnout as similar/the same. They can have different symptoms.

► There is indeed a lot of scientific debate about the definition of burn-out that we also acknowledged (see lines 338-340). We do not state that these are exactly the same, however we pose that they are both on the same continuum. While we acknowledge that burnout may have some distinctive features such as cynism, the central exhaustion dimension is very similar to distress. We indeed found that similar work-related psychosocial risk factors were associated with distress and exhaustion.

Line 340 'We used the risk estimates as reported in the original longitudinal studies (see also table 1; i.e. OR, RR and HR). Mostly they reported ORs, possibly because of the mathematical advantages of using OR'

Comment: The authors have appropriately discussed using Odds Ratios from longitudinal studies where Risk Ratios have not been provided.

► Thank you.

In addition there was an appropriate defense of using two studies as a minimum in the meta – analysis.

► Thank you.

Line 358 'For the risk factor, bullying and violence, it was decided not to perform a meta-analysis because of the different concepts of the psychosocial risk factor, i.e. person-related bullying, work-related bullying, sexual harassment, workplace bullying, threatening or violence'.

Comment: Given the concerns of the authors it would be useful to have an analysis of bullying, given that there are many types of bullying as they may share the same symptoms.

► Besides the information about measurements of bullying factors and stress related mental disorders, we added the following information about the risk factors in the results section (see lines 290-295). Four studies reported associations between bullying and SRD (Laine 2014, Taniguchi 2015, Kinda 2018, Pihl-Thingvad 2019). Laine 2014 defined workplace bullying or mental violence (1 item) to the isolation of a member of the organisation, the underestimation of work performance, being

threatened, being talked about behind one's back and other forms of pressure. Taniguchi 2017 studied person related bullying, workplace bullying and sexual harassment. Kind 2018 studied verbal and physical client aggression and Phil-Thingvad 2019 studied physical violence.

Reviewer #4

The statistical analysis appears appropriate but there are a few issues to clarify:

1. Lines 182-186: How was heterogeneity assessed?

► We used the I² for statistical heterogeneity (see line 184) and heterogeneity of risk factor and clinical outcome by discussion among the authors (see line 185).

2. Lines 194-197: Did all included studies report ORs?

► Most of them, see table 1.

3. I suggest assessing publication bias.

► Following Huguet et al. (2013) we defined no publication bias when the value of the risk factor in predicting the outcome has been repetitively investigated in explanatory studies. We also checked the funnel plots visually and found no serious indications of publication bias or could not assess this because of the very low number of included studies [undetected publication bias] for some psychosocial factors (see lines 370-374 and table 3).

4. Lines 340-342: If different measures of association were used, this should have been described in the methods (see comment 2). Are the included studies all prospective, i.e. they record exposures and follow individuals up for a period of time until outcomes are observed? Or do some have different study designs? It is not clear from the paper. They are also referred to as longitudinal.

Also it is not clear what the mathematical advantages of using ORs are.

► We now describe the different association measures in the methods (see line 117-118) and also in table 1. All studies were prospective cohort studies (see also table 3), we stated this more consistently throughout the manuscript. We skipped the sentence about mathematical advantages of using ORs, it was not relevant for this paragraph.

VERSION 2 – REVIEW

REVIEWER	Johannes Siegrist Heinrich-Heine-University Düsseldorf, Düsseldorf, Germany
REVIEW RETURNED	16-Mar-2020

GENERAL COMMENTS	This contribution provides an updated systematic review and meta-
---

	analysis of evidence on associations of adverse psychosocial working conditions and mental health outcomes based on prospective studies. The manuscript meets relevant quality criteria as it 1) covers the most important constructs of assessing stressful work, 2) justifies the selection of studies and describes studies clearly with text, tables and figures, 3) defines quality criteria of selected studies, and 4) follows GRADE in assessing quality of evidence. The text reads well, and it summarizes findings of relevance to the readership of this journal. Minor queries are as follows: 1. Literature search: Being an expert in this field of research I was wondering why the following two studies were not included in this systematic review: Wang J, Patten SB, Currie S et al. A population-based longitudinal study on work environmental factors and the risk of major depressive disorder. Am J Epidemiol. 2012;176(1):52–9; Juvani A, Oksanen T, Salo P et al. Effort-reward imbalance as a risk factor for disability pension: the Finnish Public Sector Study. Scand J Work Environ Health. 2014;40(3):266–77. 2. Description of studies and quality of evidence: In Table 3, additional information of duration of follow-up from exposure to health outcome would be relevant. Specifically, studies with low duration (e.g. below 1 year) should be excluded due to risk of reverse causation. At least, authors should discuss their lack of providing this information as a study limitation. 3. Quality of evidence: I agree with authors that this issue is still debated. Reference to Huguet et al. 2014 is okay, and risk of bias is well done. Lack of funnel plot is not a major limitation. Perhaps, in their Discussion, authors could have made more explicit that assessing strength of evidence is problematic in observational studies although the judgment of ‘moderate evidence’ seems valid. For reference see e.g. Woodruff, T.J., Sutton, P., 2014. The Navigation Guide systematic review methodology: a rigorous and transparent method for translating environmental health science into better health outcomes. Environ Health Perspect 122:1007-1014.
--	---

REVIEWER	Chris Peterson La Trobe University Australia
REVIEW RETURNED	10-Mar-2020

GENERAL COMMENTS	Work-related psychosocial risk factors for stress-related mental disorders – an updated systematic review and meta-analysis Overall there was an improvement in presentation and additions
--

	were helpful in clarifying the study Rationale for the study (Page 6) clarifies the aim and basis for the current study. Page 20 The inclusion of additional bullying studies was most useful. Page 23/4 The grading of evidence substantially improved the article and gave a clear explanation of the process used in the classification of studies.
--	--

REVIEWER	Christiana Kartsonaki University of Oxford
REVIEW RETURNED	11-Mar-2020

GENERAL COMMENTS	The authors have addressed my previous comments.
--

VERSION 2 – AUTHOR RESPONSE

Reviewer(s)' Comments to Author:

Reviewer: #2

This contribution provides an updated systematic review and meta-analysis of evidence on associations of adverse psychosocial working conditions and mental health outcomes based on prospective studies. The manuscript meets relevant quality criteria as it 1) covers the most important constructs of assessing stressful work, 2) justifies the selection of studies and describes studies clearly with text, tables and figures, 3) defines quality criteria of selected studies, and 4) follows GRADE in assessing quality of evidence. The text reads well, and it summarizes findings of relevance to the readership of this journal.

► Thank you!

Minor queries are as follows:

1. Literature search: Being an expert in this field of research I was wondering why the following two studies were not included in this systematic review:

Wang J, Patten SB, Currie S et al. A population-based longitudinal study on work environmental factors and the risk of major depressive disorder. *Am J Epidemiol.* 2012;176(1):52–9;

Juvani A, Oksanen T, Salo P et al. Effort-reward imbalance as a risk factor for disability pension: the Finnish Public Sector Study. *Scand J Work Environ Health.* 2014;40(3):266–77.

► Our review was aimed at stress related disorders covering diagnoses of adjustment disorders (ICD-10: F43.2) and burn-out as a state of exhaustion (ICD-10: Z730). The two suggested studies had depressive disorders as outcome, and therefore not eligible for our systematic review.

2. Description of studies and quality of evidence:

In Table 3, additional information of duration of follow-up from exposure to health outcome would be relevant. Specifically, studies with low duration (e.g. below 1 year) should be excluded due to risk of reverse causation. At least, authors should discuss their lack of providing this information as a study limitation.

► One of the criteria for methodological quality assessment of the individual studies was that the follow-measurements on data outcome were collected at least at three different time points, so this takes the follow-up duration into account. We acknowledged this in the results section, namely the most frequently missing quality items were: response at follow-up less 70%, data on outcome were collected at least at three different time points, and study controls for confounding with rationale. We added an extra limitation in the discussion, namely 'however, in only four out 17 prospective studies in this systematic review data on outcome were collected at least at three different time points'.

3. Quality of evidence:

I agree with authors that this issue is still debated. Reference to Hugué et al. 2014 is okay, and risk of bias is well done. Lack of funnel plot is not a major limitation. Perhaps, in their discussion, authors could have made more explicit that assessing strength of evidence is problematic in observational studies although the judgment of 'moderate evidence' seems valid. For reference see e.g. Woodruff, T.J., Sutton, P., 2014. The Navigation Guide systematic review methodology: a rigorous and transparent method for translating environmental health science into better health outcomes. *Environ Health Perspect* 122:1007-1014.

► Thank you. We feel that we have addressed that assessing strength of evidence is problematic in observational studies although the judgment of 'moderate evidence' seems valid in the discussion. In addition, we also explicitly addressed this in an extra bullet of the strength and limitations section after the abstract, namely 'The GRADE framework made it possible to assess to the quality of evidence, however, the starting level for grading this evidence for prospective etiologic studies is subject to scientific debate'.

Reviewer #3

Overall there was an improvement in presentation and additions were helpful in clarifying the study; Rationale for the study (Page 6) clarifies the aim and basis for the current study; Page 20 The inclusion of additional bullying studies was most useful; Page 23/4 The grading of evidence substantially improved the article and gave a clear explanation of the process used in the classification of studies.

► Thank you.

Reviewer: #4

The authors have addressed my previous comments.

► Thank you.

VERSION 3 - REVIEW

REVIEWER	Johannes Siegrist Centre for Health and Society Faculty of Medicine Heinrich-Heine-University Düsseldorf Germany
REVIEW RETURNED	12-May-2020
GENERAL COMMENTS	I have read the authors' answers to my queries and they have resolved the Problems. Therefore, I recommend to publish the manuscript in its updated Version.